# Black String Solutions in Rainbow Gravity

Roberta Dárlla [1], Francisco A. Brito [1] and Job Furtado [2,*]

1 Departamento de Física, Universidade Federal de Campina Grande, Campina Grande 58429-900, PB, Brazil
2 Centro de Ciência e Tecnologia, Universidade Federal do Cariri, Juazeiro do Norte 63048-080, CE, Brazil
* Correspondence: job.furtado@ufca.edu.br

**Abstract:** In this paper, we studied black string solutions under the consideration of rainbow gravity. We analytically obtained the solution for four-dimensional black strings in terms of the functions $f(E/E_p)$ and $g(E/E_p)$ that sets the energy scale where the rainbow gravity becomes relevant. We also obtained the Hawking temperature for the black string, from which we can see that the rainbow functions play the role of increasing or decreasing the Hawking temperature for a given horizon radius depending on the choice of such rainbow functions. We computed the entropy, specific heat and free energy for the black string. The entropy and specific heat exhibit a rainbow dependence, whereas the free energy is not modified by the rainbow functions. Finally, we studied the effects of rainbow gravity in the orbits of massive and massless particles around a black string. We could verify that neither massive nor massless particles exhibit stable orbits around a black string in the scenario of rainbow gravity for any configuration of rainbow functions.

**Keywords:** black string; rainbow gravity; modified dispersion relation; thermodynamics

## 1. Introduction

Black holes are obtained as solutions of Einstein's equations and play a very relevant role in physics since such objects can be used to understand how space-time is established after a gravitational collapse. Although there is a natural tendency to study spherically symmetric black holes, especially in space-times with a vanishing cosmological constant, the study of such objects with different topologies has also become something of interest. Space-times with a negative cosmological constant are the background for the existence of black holes with cylindrical symmetry. Black strings emerge in this scenario. A cylindrically symmetric black hole solution, namely a black string, for a four-dimensional Einstein–Hilbert action was proposed in [1] in the context of the classical theory of gravitation, and since then, it has received much attention in the literature.

Double special relativity (DSR) arises aiming to describe ultra-energetic particles and suggests that there is a minimum length scale, which implies a maximum energy scale—the Planck scale—so that the speed of light is not the only relativistic invariant and that there is an energy scale (or length) that is independent of the observer. In such models, the dispersion relation is modified when we consider energies near to the Planck scale [2–5].

The approaches that receive the name of rainbow's gravity emerge in this scenario and seek to propose a modification to the classical theory of general relativity by incorporating aspects of the theory of modified special relativity (DSR). According to this theory, there is no single classical geometry for the space-time when the energy scale approaches the Planck scale. These approaches have been studied in several scenarios, such as string theory [6], loop quantum gravity [7] and non-commutative geometry [8]. Some theoretical proposals suggest corrections both in the action and in the dispersion relation, such as in [9].

This semi-classical approach allows some phenomena to be explained, such as ultra-high-energy cosmic rays that are currently observed but still have unknown origin, suggesting that the dispersion relation is indeed modified. In astrophysics, the influence of the

rainbow's gravity on the properties of a black hole has been studied in several scenarios, including its thermodynamics [10–20], and also in the study of cosmic strings [21–23]. In addition, in order to understand the early universe, in which the energies involved were close to the Planck scale, such a modified theory of gravity plays an important role in avoiding an initial singularity [24–29]. Finally, in general field theory, there have been many recent developments regarding rainbow gravity in the context of Bose–Einstein condensation [30], Klein–Gordon oscillation [29], the Landau–Aharonov–Casher effect [31] and particle production [32], among others.

In this paper, we studied black string solutions under the consideration of rainbow gravity. We analytically obtained the solution for four-dimensional black strings in terms of the functions $f(E/E_p)$ and $g(E/E_p)$ that sets the energy scale where the rainbow gravity becomes relevant. We also obtained the Hawking temperature and other thermodynamic quantities for the black string. We also investigated the orbits for massive and massless particles.

This paper is organized as follows. In Section 2, we present a brief review of rainbow gravity in which we present the most common ansatz for the rainbow functions investigated in the literature. In Section 3, we obtain the black string solution in the context of rainbow gravity while, in Section 4, we investigate the black string thermodynamics under the influence of rainbow gravity. The orbits for massive and massless particles are studied in Section 5 and, in Section 6, we present our conclusions.

## 2. Rainbow Gravity Review

Rainbow gravity was first proposed over a decade ago, it being studied within the scope of double special relativity (DSR). This approach emerges assuming that there is no single classical geometry for spacetime; however, the geometry may depend on the energy of a particle moving in it. This means that particles with different energies distort spacetime differently. One result that follows from this is the emergence of a modified energy–momentum dispersion relation. Such a modification is usually written in the form [4,33,34]

$$E^2 f^2(E/E_P) - p^2 c^2 g^2(E/E_P) = m^2 c^4, \tag{1}$$

where $f(E/E_P)$ and $g(E/E_P)$ receive the generic name of rainbow functions, and these functions are parameterized by the ratio $E/E_P$, where $E$ is the total energy of the particle or system of particles measured by a free falling observer and $E_P$ is the energy on the Planck scale. These functions are constructed so that, in the low energy limit, they converge to a unit.

Since the spacetime geometry changes according to the energy in the test particle in it, there is no single spacetime dual to the momentum space, i.e., there is a family of energy-dependent metrics, according to [34], that will be parameterized by the rainbow functions. The construction of the metric must be carried out in such a way that it is covariant in relation to the non-linear representation of the Lorentz transformations; thus, the Minkowski spacetime becomes

$$ds^2 = \frac{dt^2}{f^2(E/E_P)} - \frac{1}{g^2(E/E_P)} \delta_{ij} dx^i dx^j. \tag{2}$$

In order to study the rainbow gravity effects on the Friedmann–Robertson–Walker (FRW) universe [12,24], the following rainbow functions were considered (case I):

$$f(E/E_P) = 1, \quad g(E/E_P) = \sqrt{1 - \xi(E/E_P)^s}, \tag{3}$$

where $s > 1$ and $\xi$ is a dimensionless free parameter of the model that we will consider the same as the other rainbow functions to facilitate comparison between the employed models.

Another interesting choice for the rainbow functions is the following (case II):

$$f(E/E_P) = g(E/E_P) = \frac{1}{1 - \xi(E/E_P)}. \tag{4}$$

Such rainbow functions were considered in [4,33] (and references therein) in studying possible nonsingular universe solutions, and also in [34]. Since it assures a constant light velocity, it may provide a solution for the horizon problem.

A last choice of rainbow functions of great interest is given by (case III)

$$f(E/E_P) = \frac{e^{\xi(E/E_P)} - 1}{\xi(E/E_P)}, \quad g(E/E_P) = 1. \tag{5}$$

This choice of rainbow functions was originally considered in [35] in the context of gamma ray bursts. Later, this same choice was also addressed in [24,36] in connection with FRW solutions.

### 3. Black String Solution in Rainbow Gravity

First, let us review the black string solution for Einstein's equations with a negative cosmological constant obtained by Lemos in [1] within the scenario of the classical theory of gravitation. Let us consider the following line element:

$$ds^2 = -A(r)dt^2 + \frac{dr^2}{A(r)} + r^2 d\phi^2 + \alpha^2 r^2 dz^2, \tag{6}$$

where $-\infty < t < \infty$ and $0 \leq r < \infty$, $0 \leq \phi \leq 2\pi$ and $-\infty < z < \infty$ define the radial, angular and axial coordinates, respectively. The parameter $\alpha$ is such that $\alpha^2 \equiv -\Lambda/3 > 0$, where $\Lambda$ is the cosmological constant. For the black string, the Einstein–Hilbert effective action requires the cosmological constant contribution; thus,

$$S_u = \frac{1}{2\kappa^2} \int d^4x \sqrt{-g}(R - 2\Lambda). \tag{7}$$

where $\kappa = 8\pi G$ and $R$ is the Ricci scalar. We must use the metric given in (6) to calculate the components of the Einstein tensor, where, in our case, we only need the component $G^0{}_0$, from which we obtain the following expression:

$$G^0{}_0 = \frac{1}{r^2} \frac{d}{dr}[rA(r)], \tag{8}$$

Then, we can solve Einstein's equations, which, for this case, will be adapted to

$$G^\mu_\nu + \delta^\mu_\nu \Lambda = 8\pi T^\mu_\nu. \tag{9}$$

Assuming that $T^0_0 = 0$ to $r \neq 0$, we obtain

$$\frac{1}{r^2} \frac{d}{dr}[rA(r)] + \Lambda = 0. \tag{10}$$

Solving the above differential equation, we obtain the following usual solution defined in [1]:

$$A(r) = \alpha^2 r^2 - \frac{4\mu}{\alpha r}, \tag{11}$$

where $\mu$ is the linear mass of the black string. The event horizon can be determined by performing $A(r) = 0$, whose solution is

$$r \equiv r_h = \frac{(4\mu)^{1/3}}{\alpha}. \tag{12}$$

It is important to note that there is a singularity at $\alpha r = 0$ and that, when $r \to \infty$, the spacetime of the black string is anti-de-Sitter.

Let us now consider the rainbow gravity effects so that we have the following line element for the black string:

$$
\begin{aligned}
ds^2 \;=\; & -\frac{A(r)}{f(E/E_p)}dt^2 + \frac{1}{g(E/E_p)A(r)}dr^2 + \frac{r^2}{g(E/E_p)}d\phi^2 \\
& + \frac{\alpha^2 r^2}{g(E/E_p)}dz^2,
\end{aligned}
\tag{13}
$$

The non-vanishing components of the Einstein tensor for the black string in rainbow gravity are:

$$
G_t^t \;=\; G_r^r = \frac{g(E/E_p)^2(rA'(r) + A(r))}{r^2}
\tag{14}
$$

$$
G_\phi^\phi \;=\; G_z^z = \frac{g(E/E_p)^2(rA''(r) + 2A'(r))}{2r}
\tag{15}
$$

Hence, the EFE for this ansatz gives us

$$
G_t^t - 3\alpha^2 \;=\; [g(E/E_p)]^2\left[\frac{1}{r}\frac{dA(r)}{dr} + \frac{A(r)}{r^2}\right] - 3\alpha^2,
\tag{16}
$$

We can see that the energy–momentum tensor for the ansatz of Equation (13) is $T_\nu^\mu = -\rho(r)\,\mathrm{diag}(1,1,0,0) + p_l(r)\,\mathrm{diag}(0,0,1,1)$, where $p_l = p_\phi = p_z = (2\pi r)^{-1}\delta(r)$. This way, we can find $A(r)$ by solving $G_t^t - 3\alpha^2 = -\kappa^2\rho(r)$, where $\rho(r) = \mu(2\pi r)^{-1}\delta(r)$, so that we find

$$
A(r) = \frac{\alpha^2 r^2}{[g(E/E_p)]^2} - \frac{4\mu}{\alpha r}.
\tag{17}
$$

The above solution for the black string in the rainbow gravity scenario recovers the usual black string solution (11) when $g(E/E_p) = 1$.

Notice also that the rainbow gravity modification does not affect the regularity of the solution since the rainbow functions do not explicitly depend on the radius. The energy conditions are also unaffected by the rainbow gravity.

The behavior of the black string solution in rainbow gravity is depicted in Figure 1 for cases I and II. Note that case III for the rainbow functions does not give us any modification in the black string solution since $g(E/E_p) = 1$.

Let us briefly discuss the role played by the rainbow gravity scenario in the black string solution. As we can see in Figure 1, as we increase the value of the energy $E$, approaching the Planck energy scale, we also increase the value of the horizon radius for cases I and II of the rainbow functions.

In order to analyze the conical defect due to the black string in the rainbow scenario, let us power expand the solution (11) around $r_\alpha = 1/\alpha$; that is,

$$
A(r) = 1 - 4\mu + \mathcal{O}\left(r - \frac{1}{\alpha}\right).
\tag{18}
$$

By using (12), we can ensure that $r_\alpha > r_h$ as long as $4\mu < 1$. Thus, we can now write the line element (13) at a leading order as

$$
\begin{aligned}
ds^2 \;=\; & -\frac{(1-4\mu)dt^2}{f(E/E_p)} + \frac{dr^2}{g(E/E_p)(1-4\mu)} + \frac{r^2 d\phi^2}{g(E/E_p)} \\
& + \frac{dz^2}{g(E/E_p)}.
\end{aligned}
\tag{19}
$$

Now, applying the rescalings $ds^2/(1-4\mu) \to d\tilde{s}^2$, $dt^2 \to d\tilde{t}^2$ $dr^2/(1-4\mu)^2 \to d\tilde{r}^2$, $d\phi^2/(1-4\mu) \to d\tilde{\phi}^2$ and $dz^2/(1-4\mu) \to d\tilde{z}^2$, we find the metric of a conical defect given by

$$
\begin{aligned}
d\tilde{s}^2 \;=\; & -\frac{d\tilde{t}^2}{f(E/E_p)} + \frac{d\tilde{r}^2}{g(E/E_p)} + \frac{(1-4\mu)^2 \tilde{r}^2 d\tilde{\phi}^2}{g(E/E_p)} \\
& + \frac{d\tilde{z}^2}{g(E/E_p)}.
\end{aligned}
\tag{20}
$$

At the limit of $f(E/E_p) = g(E/E_p) = 1$, we recover the well-known metric due to a cosmic string in the Minkowski form with a deficit angle of $\Delta\tilde{\phi} = 8\pi\mu$ in the regime of $\mu \ll 1$; that is, a solution obtained from the Einstein equations in a weak gravitational field approximation for the energy–momentum tensor [37]

$$
T_\mu^\nu = \mu\,\delta(x)\delta(y)\,\mathrm{diag}(1,0,0,1).
\tag{21}
$$

Furthermore, the limit $\mu \ll 1$ implies that $r_\alpha \gg r_h$, which is also consistent with $\alpha \ll 1$.

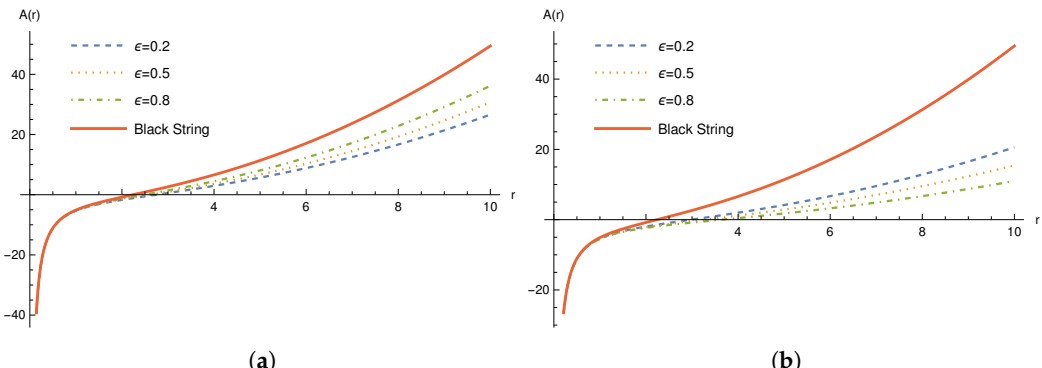

(**a**)  (**b**)

**Figure 1.** Black string solution in rainbow gravity. For this plot, we consider $E_p = 1$, $s = 1$, $\xi = 0.4$, $\alpha = 0.5$ and $\mu = 0.7$. In (**a**), we consider case I for the rainbow functions whereas, in (**b**), we consider case II.

## 4. Black String Thermodynamics in Rainbow Gravity

The second law of black holes states that, for any physical process, the surface area $A$ of a black hole's event horizon obeys the relation $\delta A \geq 0$, where $\delta A = 0$ for stationary processes, and thus the increase in the area of the event horizon is caused by non-stationary processes. In an analogous way, the entropy of a system of particles, defined by the second law of thermodynamics, follows the relation $\delta S \geq 0$, with $\delta S = 0$ for reversible processes. Although this seems just a mere coincidence since the second law for black holes comes from general relativity while entropy is a consequence of the fact that a physical system has many degrees of freedom, it is possible to find a parallel between all the laws of thermodynamics and the laws that black holes satisfy, showing that this relation is fundamental and not just a coincidence [38].

Our black string solution in the rainbow gravity scenario has horizon curves defined by $A(\tilde{r}_h) = 0$; thus, the linear mass can be written as

$$
\mu = \frac{\alpha^3 \tilde{r}_h^3}{4g(E/E_p)^2}.
\tag{22}
$$

Here, $\tilde{r}_h = r_h\,[g(E/E_P)]^{2/3}$, where $r_h$ is the horizon radius of the usual general relativity solution for the black string. Then, the expression (22) becomes

$$
\mu = \frac{\alpha^3 r_h^3}{4}.
\tag{23}
$$

Thus, this linear mass has no modification due to the rainbow gravity.

In possession of the solution for the static black string in the rainbow gravity scenario given by (17), we are able to study the thermodynamics of the black string by computing the Hawking's temperature by means of $T_H = \frac{A'(\tilde{r}_h)}{4\pi}$. Thus, we obtain

$$\tilde{T}_H = \frac{3\alpha^2 \, r_h}{4\pi \, [g(E/E_P)]^{4/3}}. \tag{24}$$

The behavior of the Hawking temperature for cases I and II is depicted in Figure 2. For both cases (I and II), the same linear behavior of the usual Hawking temperature for black strings is present. However, some slight differences between the cases must be highlighted. For case I (Figure 2a), we can see that, for a given horizon radius, the Hawking temperature is greater when we consider the effect of rainbow gravity. The opposite occurs for case II (Figure 2a), where, for a given horizon radius, the Hawking temperature is smaller when we consider the effect of rainbow gravity.

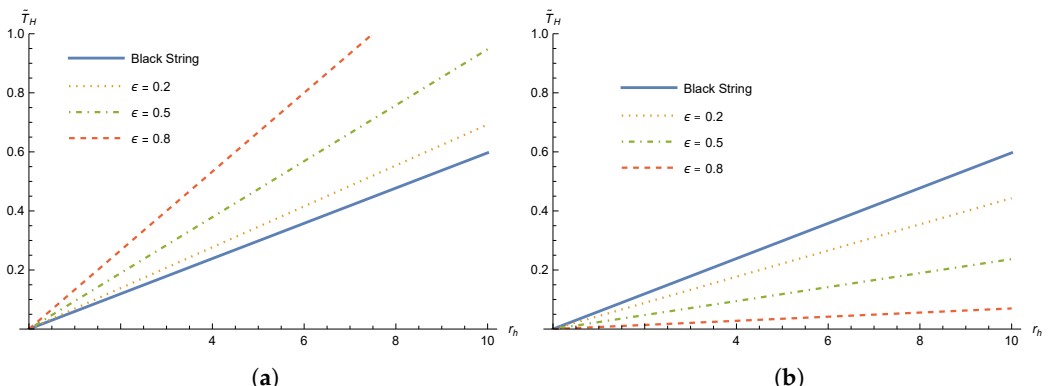

(a)                                   (b)

**Figure 2.** Hawking temperature for black string solution in rainbow gravity. For this plot, we consider $E_p = 1$, $s = 1$, $\xi = 0.4$, $\alpha = 0.5$ and $\mu = 0.7$. In (**a**), we consider case I for the rainbow functions whereas, in (**b**), we consider case II.

In order to properly understand the thermodynamics of the black string in the rainbow gravity context, it is necessary to compute the entropy, specific heat and free energy. The entropy can be computed directly from the expression $dS = \frac{d\mu}{\tilde{T}_H}$, in which we obtain

$$\tilde{S} = \frac{\pi \, \alpha \, r_h^2 \, [g(E/E_P)]^{4/3}}{2}. \tag{25}$$

Clearly, this recovers the usual black string result $S = \frac{1}{2}\pi\alpha r_h^2$ when $g(E/E_P) = 1$. As we can see in Figure 3, for both cases, we have the same quadratic dependence of the horizon radius that the usual black string entropy exhibits. However, differently from the Hawking temperature, case I promotes a decrease in the entropy for a given horizon radius whereas case II promotes an increase in the entropy for a given horizon radius.

The specific heat can be calculated by $\tilde{C}_v = \frac{d\mu}{d\tilde{T}_H}$, from which we obtain

$$\tilde{C}_v = \pi\alpha r_h^2 \, [g(E/E_P)]^{4/3} \tag{26}$$

Similar to entropy, in case I, for a given horizon radius, the specific heat is smaller when we consider the effect of rainbow gravity. The opposite happens for case II. When $g(E/E_P) = 1$, we obtain $C_v = \pi\alpha r_h^2$, i.e., the usual black string specific heat in general relativity. The behavior of the specific heat for cases I and II of the rainbow functions is depicted in Figure 4. As is widely known, the thermodynamic stability of black holes (black strings for our case) is directly related to the sign of the heat capacity. A positive heat capacity indicates that the system is thermodynamically stable, whereas its negativity

implies thermodynamic instability. Therefore, the result for the specific heat in the context of rainbow gravity indicates a thermodynamically stable black string.

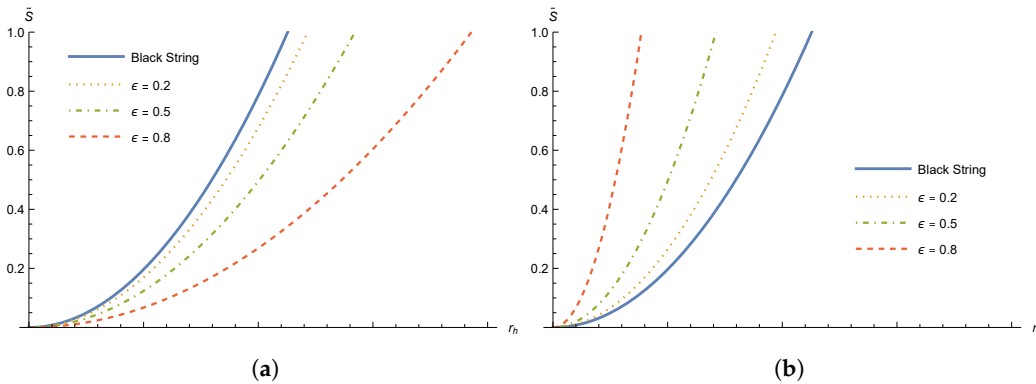

**Figure 3.** Entropy for black string solution in rainbow gravity. For this plot, we consider $E_p = 1$, $s = 1$, $\xi = 0.4$, $\alpha = 0.5$ and $\mu = 0.7$. In (**a**), we consider case I for the rainbow functions whereas, in (**b**), we consider case II.

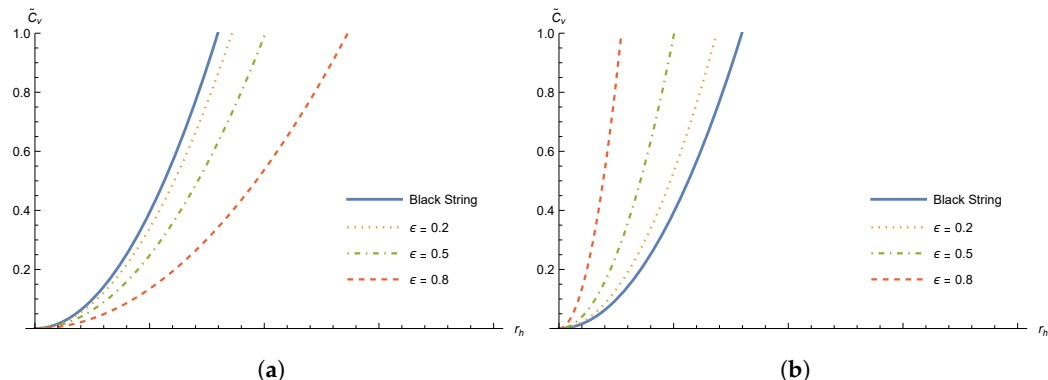

**Figure 4.** Specific heat for black string solution in rainbow gravity. For this plot, we consider $E_p = 1$, $s = 1$, $\xi = 0.4$, $\alpha = 0.5$ and $\mu = 0.7$. In (**a**), we consider case I for the rainbow functions whereas, in (**b**), we consider case II.

On the other hand, the rainbow gravity presents no modification in the free energy $F = \mu - T_H S$, therefore yielding the usual black string result

$$F = -\frac{\alpha^3 r_h^3}{8}. \tag{27}$$

## 5. Geodesics and Circular Orbits

Another important result that we can obtain is the possible circular orbits for this black string solution. We can investigate such orbits through the effective potential energy ($V_r$) of a system formed by the black string and a massive particle and a system formed by the black string and a photon. A stable orbit is defined as one whose $V_{eff}'' > 0$; that is, in the vicinity of the equilibrium point, the curve's concavity is up, whereas an unstable orbit has $V_r'' < 0$; that is, in the vicinity of the equilibrium point, the curve's concavity is down [39].

The particle's geodesic in orbit around a static black string is given by

$$\dot{r}^2 = \omega^2 - A(r)\left(\frac{L^2}{r^2} + m^2\right), \tag{28}$$

where $\omega$ is the particle's energy, $L$ is the angular momentum and $m$ is the particle's mass. Thus, the effective potential is defined as

$$V_r = A(r)\left(\frac{L^2}{r^2} + m^2\right). \tag{29}$$

The circular geodesics occur at the points $r_c$ satisfying $\frac{1}{2}\dot{r}_c^2 = 0$ and $V_r'(r_c) = 0$. In Figure 5, we depict the effective potential of massless and massive particles for the black string in the rainbow gravity scenario. It is shown that there is no case where circular orbits are stable, similarly to the usual black string solution. Therefore, the rainbow gravity does not significantly modify the results for geodesics and circular orbits in comparison to the usual black string.

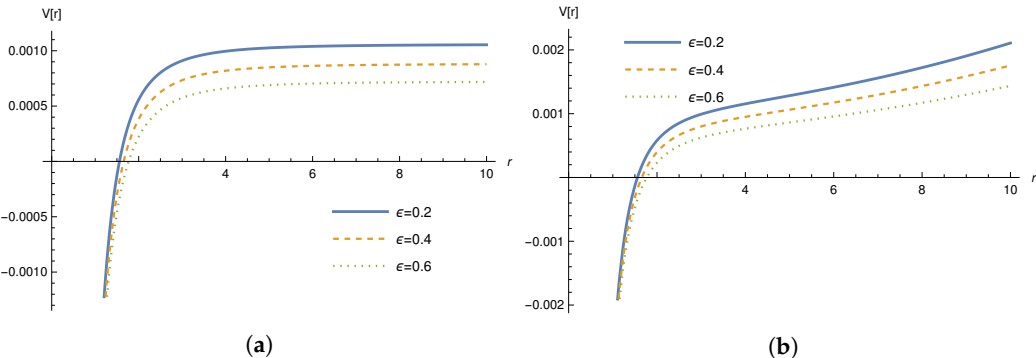

(**a**)                                                                                              (**b**)

**Figure 5.** Effective potential (**a**) for massless particles and (**b**) for massive particles. For this plot, we consider $E_p = 1$, $s = 1$, $\xi = 0.4$, $\alpha = 0.5$, $\mu = 0.7$ and $L = 0.1$.

## 6. Conclusions

In this paper, we studied black string solutions under the consideration of rainbow gravity. We analytically obtained the solution for four-dimensional black strings in terms of the functions $f(E/E_p)$ and $g(E/E_p)$ that sets the energy scale where the rainbow gravity becomes relevant. We could verify that the black string solution depends only on the function $g(E/E_p)$ and, consequently, that all the thermodynamic parameters will depend only on $g(E/E_p)$. We plotted the behavior of the black string solution in (Figure 1) and could see that, as we increase the value of the energy $E$, approaching the Planck energy scale, we also increase the value of the horizon radius for cases I and II of the rainbow functions.

We also obtained the Hawking temperature for the black string, from which we could see that the rainbow functions play the role of increasing or decreasing the Hawking temperature for a given horizon radius depending on the choice of such rainbow functions. We computed the entropy, specific heat and free energy for the black string. The entropy and specific heat exhibit a rainbow dependence, whereas the free energy is not modified by the rainbow functions.

Finally, we studied the effects of the rainbow gravity in the orbits of massive and massless particles around a black string. We could verify that neither massive nor massless particles exhibit stable orbits around a black string in the scenario of rainbow gravity for any configuration of rainbow functions.

**Author Contributions:** Conceptualization, R.D., F.A.B. and J.F.; investigation, R.D., F.A.B. and J.F.; writing—original draft, R.D., F.A.B. and J.F.; writing—review and editing, R.D., F.A.B. and J.F. All authors have read and agreed to the published version of the manuscript.

**Funding:** CNPq 309092/2022-1 and FUNCAP PRONEM PNE0112-00085.01.00/16.

**Data Availability Statement:** All data are available in the paper.

**Acknowledgments:** F.A.B. would like to thank CNPq and CNPq/PRONEX/FAPESQ-PB (Grant nos. 165/2018 and 309092/2022-1) for partial financial support. J.F. would like to thank the Fundação Cearense de Apoio ao Desenvolvimento Científico e Tecnológico (FUNCAP) under the grant PRONEM PNE0112-00085.01.00/16 for financial support.

**Conflicts of Interest:** There is no conflict of interest regarding this research.

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
