# Peer review of "Black String Solutions in Rainbow Gravity"

_universe, doi:10.3390/universe9060297_

Round 1
Reviewer 1 Report
In this paper they study black string solutions under the consideration of rainbow gravity. They have analytically obtained the solution for four-dimensional black strings in terms of the functions f (E=Ep) and g(E=Ep) that sets the energy scale where the rainbow gravity becomes relevant. They have also obtained the Hawking temperature for the black string, from which could see that the rainbow functions play the role of increasing or decreasing the Hawking temperature for a given horizon radius depending on the choice of such rainbow functions. They have computed the entropy, specific heat and free energy for the black string. The entropy and specific heat exhibit a rainbow dependence, while the free energy is not modified by the rainbow functions. Finally they have studied the effects of the rainbow gravity in the orbits of massive and massless particles around a black string.The paper raises the interesting and actual questions of the possibilities to black string solutions under the consideration of rainbow gravity. It is good structured and written in a clearly understandable way. I think it can be published in its present form.
Reviewer 2 Report
This is an article on the polemic rainbow gravity theory. That being said, the article itself has its merits and its problems. The article is basically a well-structured classical analysis of a set of geometries interconnected by changes in a set of functions of energy.
Let me address the problems now. First, the lesser problem. The article never mentioned how the solutions were obtained. This is very important as any traditional analysis would require knowing an action principle or a set of equations, to, for instance, determine the mass and other conserved charges. This traditionally would be of utmost relevance given that without those charges the construction of a first law of thermodynamics would be impossible.
The biggest issue, however, is something of a more fundamental nature, which I must admit is obviously far beyond the scope of this work. In the context of rainbow gravity, where it is to be considered that each energy defines a different trajectory, then the very definition of black-body radiation must be changed to incorporate the splitting of trajectories, in spacetime, as a function of the frequencies/energies. This, unfortunately, spoils the connection between black hole physics with thermodynamics. It is good to remain that the temperature of a black hole, in usual gravity, has several different interpretations that only unify under the idea of a single definition, for any observer, of a geodesic incomplete region that one can identify with the exterior of the black hole. Even in the simplest case, only under that assumption, the "temperature of the radiation" of particles emitted by that region can be interchanged with the would-be Euclidean period defined by the vanishing of the null (Killing) generator of the horizon. In other words, beyond the details of the analysis performed, doing the analysis itself, and in the way presented, is flawed. The very idea of any set of laws of thermodynamics that changes its form as a function of the energy changes lacks any meaning.
Therefore, I regret to comment that without a clarification of these previous points, it is impossible for me to recommend the publication of this article.
Reviewer 3 Report
The paper is devoted to the studies of black string in the so-called rainbow gravity. The authors obtained the metric of the object in terms of fractions of energy scales. They calculated and graphically presented the dependence of the metric function and Hawking temperature, on the energy ratios. The effects of rainbow gravity on massive and massless particles orbits in the victim of black string, have been also considered.
In my opinion the work is interesting and will attract attention of the people working in the subject. Therefore I recommend it for publication.
Round 2
Reviewer 2 Report
After reviewing the new version of the text I must admit that my doubts about the material presented not only continue, they have increased.
My main objection is not about the work itself, but about the whole framework. In my opinion, the presentation is adequate, however, the solution displayed, and the way the equations of motion are solved, which are sound within Rainbow gravity, are completely flawed. For instance, what is the matter that generates the conical defect? There is a natural contradiction between the usual framework of GR and the equations presented in this work. In GR the conical defect is generated by a cosmic string with a well-defined equation of motion in the geometry. Here, on the other hand, the corresponding equations of motion of the string are undefined. Furthermore, which would they be in a framework where the dynamics depends on the energy scale? This requires further clarification in my opinion before considering publication.
